A new species of long-necked turtle (Pleurodira: Chelidae: Chelodina) from the late Miocene Alcoota Local Fauna, Northern Territory, Australia

Yates Adam M. adamm.yates@nt.gov.au
Museums and Art Galleries of the Northern Territory, Museum of Central Australia , Alice Springs, Northern Territory , Australia
Farke Andrew
Electronic publication date: 2013 Oct 1
Publication date: 2013
Volume: 1
Electronic Location ID: e170
Received 2013 Jun 20; Accepted 2013 Sep 5
Copyright: © 2013 Yates
Copyright year: 2013
Copyright holder: Yates
License: This is an open access article distributed under the terms of the Creative Commons Attribution License, which permits unrestricted use, distribution, and reproduction in any medium, provided the original author and source are credited.
License URL: https://creativecommons.org/licenses/by/3.0/

Keywords: Chelidae, Miocene, Alcoota, Waite Formation, Chelodina, Australia

Funding: The author received no funding for this study.

==============================
The new species Chelodina (Chelodina) murrayi is described from the late Miocene Alcoota Local Fauna of central Australia, in the Northern Territory. The new species is based on shell fragments and can be diagnosed by a ventrally reflexed anterior margin of the plastron, a ventrally narrowed cervical scute and strongly dorsally curved margins of the carapace extending from approximately peripheral two to peripheral nine or ten as well as by a unique combination of characters. Within Chelodina the new species is part of the nominal subgenus and within that subgenus it is most closely related to the Chelodina (Chelodina) novaeguineae species group. This is not only the oldest record but also the most southerly occurrence of this species group.

Introduction

Chelodina is an extant genus of chelid turtle found in Australia, New Guinea, East Timor and Roti Island, Indonesia (Georges & Thomson, 2010). Popularly known as long-necked or snake-necked turtles they are readily distinguished from other Australasian chelids by their elongated necks, four-clawed forelimbs and contacting gular scutes on the plastron (Cogger, 1975; Georges & Thomson, 2010). The genus is also distinguished by a distinctive skull morphology including the loss of the temporal bar, fusion of the frontals, nasals separated by an anterior process of the frontals and an extensive quadrate-basisphenoid contact (Gaffney, 1977). Gaffney (1977) proposed that the genus shares closer relationships with long-necked South American chelids (such as Hydromedusa and Chelus) than with short-necked Australasian chelids on the basis of shared derived morphological features. However phylogenetic analyses of nuclear and mitochondrial genes indicate that reciprocal monophyly of South American and Australasian chelids is a much more likely scenario (Seddon et al., 1997; Georges et al., 1998). In keeping with its probable monophyly, Georges et al. (1998) unwittingly resurrected the old name Chelodininae Baur, 1893 for the Australasian clade when they proposed the same name as new. Within Chelodininae, Chelodina is invariably the sister taxon of all other Australasian chelids in all analyses where Chelodininae is recovered, or the taxon sampling is restricted to chelodinines (Georges & Adams, 1992; Seddon et al., 1997; Georges et al., 1998; Thomson & Georges, 2009).

Within Chelodina three separate species groups can be recognised on the basis of morphology and electrophoretic analysis of allozymes (Burbidge, Kirsch & Main, 1974; Georges, Adams & McCord, 2002). These groups have been assigned separate generic names (Wells & Wellington, 1985; McCord & Ouni, 2007) but a recent synthetic work on Australian turtle taxonomy has treated them as subgenera (Georges & Thomson, 2010) and this practice is followed here. C. (Chelodina) includes smaller species with broad plastron and a slender head and neck with a length that does not exceed that of the carapace (Georges & Thomson, 2010). C. (Macrochelodina) includes larger species with a long broad head and neck that exceeds the length of the carapace and a narrow plastron. C. (Macrodiremys) includes a single species, C. (Macrodiremys) colliei, which is outwardly similar to C. (Macrochelodina) but has some unusual characters such as a continuous row of exposed neural bones in the carapace and a lack of posterior expansion of the carapace (Burbidge, Kirsch & Main, 1974; Georges & Thomson, 2010). Furthermore molecular phylogenetics has consistently found C. (Macrodiremys) colliei to be more closely related to C. (Chelodina) than to C. (Macrochelodina), supporting its separation from the latter (Seddon et al., 1997; Georges et al., 1998; Georges, Adams & McCord, 2002).

The Fossil Record of Chelodina

Chelodina has a long fossil record, as one might expect from such a deep branch within Chelodininae. However it is much sparser than the fossil record of the short-necked chelodinine clade.

The earliest specimens referred to Chelodina have been found in the Eocene deposits of Redbank Plains in south eastern Queensland. Here there are three forms known from incomplete shells, one of which has been named C. alanrixi (Lapparent de Broin & Molnar, 2001). None of these forms can be easily assigned to the extant subgenera, not least because the material is limited and incomplete. In the case of C. alanrixi there are some similarities with C. (Macrochelodina) but there are also some notable plesiomorphies such as a complete row of exposed neural bones in the carapace and a posterior vertebral scute that is as wide as the suprapygal scute (Lapparent de Broin & Molnar, 2001). These indicate that C. alanrixi could well belong to the Chelodina stem-group, rather than to any of the recognised subgenera but this will need to be tested with more complete fossils and a phylogenetic analysis of Chelodina ingroup relationships that includes fossils.

There is a considerable gap between the Eocene and the next oldest occurrences of Chelodina in the early to middle Miocene carbonate deposits of Riversleigh in Queensland and Bullock Creek in the Northern Territory. Chelodina has been recorded from two sites at Riversleigh, Gag Site and Quentin’s Quarry, both of which have been assigned to system C (Archer et al., 1989). However, constrained seriation analysis indicates that Quentin’s Quarry is referrable to the early Miocene Wipajirian Australian Land Mammal Age, whereas Gag Site belongs to the younger middle Miocene Camfieldian ALMA (Megirian et al., 2010). The Quentin’s Quarry specimen is an intriguing skull fragment that exhibits several distinct autapomorphies and cannot be easily placed in any of the extant subgenera (White, 1997). The specimen from Gag site is a fragmentary shell that would appear to have affinities C. (Macrochelodina) based on the absence of exposed neural bones and the length of its intergular scute which is only slightly longer than the midline contact of the pectoral scutes (Gaffney, Archer & White, 1989).

The Camfield beds of Bullock Creek are Camfieldian in age and have produced two distinct Chelodina species. Both species are only known with certainty from isolated epiplastra (Megirian & Murray, 1999). One of these, Chelodina sp. B, may belong to C. (Chelodina) based on the squared-off profile of the anterior lobe of the plastron.

The record becomes a little better in the Plio-Pleistocene. From Tara Creek in Queensland there is a Chelodina specimen that is likely to be similar in age to the nearby Bluff Downs local fauna (Gaffney, 1981). The Bluff Downs local fauna is Early Pliocene in age and belongs to the Tirarian ALMA (Megirian et al., 2010). The specimen is unusual in that it lacks a finely ornamented surface even though the seams between scutes are well-marked (Gaffney, 1981). Based on the narrow anterior lobe of the plastron, a long inter-pectoral seam which exceeds the length of the intergular scute (Gaffney, 1981, Fig. 8), this specimen would appear to be a member of Chelodina (Macrochelodina).

From the Pliocene Bluff downs local fauna itself there is a single nuchal bone that has been identified as Chelodina based largely on the presence of a broad, square-shaped cervical scute (Thomson & Mackness, 1999). The specimen was tentatively assigned to the nominate subgenus (as the ‘Chelodina longicollis group’) on the basis of its well developed ornamentation. However, as the authors themselves note, this is a variable character, for example Chelodina (Macrochelodina) insculpta also has well-developed surficial ornamentation (Thomson, 2000, Fig. 4).

Chelodina (Macrochelodina) insculpta is a named fossil species based on fragments from the Plio-Pleistocene of the Darling Downs that bear a close resemblance to C. (M.) expansa but differ from it in a having a less flared margin of the carapace (Thomson, 2000).

Lapparent de Broin & Molnar (2001) point out that a posterior plastral fragment figured by Gaffney (1981, Fig. 18A) belongs to Chelodina based on the rounded shape of the posterolateral margin of the ischiadic scar on the dorsal surface of the xiphiplastron and the rounded nature of the fine tubercles that decorate its ventral surface. This fragment comes from the late Pleistocene, or Naracourtean, Katapiri Formation of Lake Kanunka, in the Tirari Desert of central South Australia. A second probable Pleistocene from central South Australia is represented by an isolated cervical vertebra from Cooper Creek (Gaffney, 1981). Unfortunately neither specimen displays diagnostic characters that would allow determination of their subgeneric affinities. Nonetheless these specimens are interesting because they show that Chelodina was surviving in central Australia until quite recently, well outside its present range.

Lastly there is a Naracourtean (late Pleistocene) record of Chelodina from Henshkes Cave, in south eastern South Australia based on isolated plastron elements (Gaffney, 1981). These are similar to the extant C. (Macrochelodina) expansa (Gaffney, 1981).

While it is highly likely that remains of the widespread, common species C. (Chelodina) longicollis are present in some of many Naracourtean vertebrate faunas of south eastern Australia, none have been positively recorded.

Thus C. (Chelodina) has an exceptionally poor fossil record, if any at all, with just a couple of fragmentary possible occurrences reported in the literature. Here I describe a new species of this subgenus from the late Miocene vertebrate fossil locality of Alcoota Station, Northern Territory. This is the first definite occurrence of the subgenus in the fossil record and the first diagnosable extinct species in the subgenus.

The electronic version of this article in Portable Document Format (PDF) will represent a published work according to the International Commission on Zoological Nomenclature (ICZN), and hence the new names contained in the electronic version are effectively published under that Code from the electronic edition alone. This published work and the nomenclatural acts it contains have been registered in ZooBank, the online registration system for the ICZN. The ZooBank LSIDs (Life Science Identifiers) can be resolved and the associated information viewed through any standard web browser by appending the LSID to the prefix “http://zoobank.org/”. The LSID for this publication is: Zoobank urn:lsid:zoobank.org:act:9DDFE14D-8E52-4F7D-B148-F552100865C9. The online version of this work is archived and available from the following digital repositories: PeerJ, PubMed Central and CLOCKSS.

Geological Setting

The Waite Formation is a late Neogene filling of the Waite Basin, a small intracratonic basin located in the Northern Territory to the northeast of Alice Springs (Woodburne, 1967). It is a coarsening-upward sequence of fluviatile silts and sands with minor limestone beds (Woodburne, 1967). Two vertebrate fossil-bearing horizons are known, one giving rise to the older Alcoota local fauna and the other the younger Ongeva local fauna (Woodburne, 1967; Megirian, Murray & Wells, 1996). The Alcoota Local Fauna is found in an extensive, dense, jumbled bone bed. The bone bed is thought to have formed from a debris flow that incorporated the skeletons of many hundreds of animals that had died in a drought-related mass mortality event around a major waterhole in an ancient river system (Murray & Vickers-Rich, 2004). Almost all of the bones in the deposit are disarticulated and randomly scattered. The bulk of the fossils belong to terrestrial species, dominated by the small macropodine wallaby, Dorcopsoides, several diprotodontid marsupials and large dromornithid birds (Murray & Megirian, 1992). Turtles are rare in this deposit and are only known from isolated bones of the shell. These fragments have received little attention in the scientific literature. They were first mentioned by Newsome & Rochow (1964) and then again by Gaffney (1981). Neither were able to identify the limited material as anything more precise than ‘Testudines indeterminate’. An intensive collecting program led by the Museum and Art Gallery of the Northern Territory over the past three decades has produced many more pieces of turtle shell, including diagnostic elements that allow a new species to be recognised.

Systematic Palaeontology

TESTUDINES Linnaeus, 1758

PLEURODIRA Cope, 1864

CHELIDAE Gray, 1831

CHELODININAE Baur, 1893

Chelodina Fitzinger, 1826

Chelodina (Chelodina) (Fitzinger, 1826)

Chelodina (Chelodina) murrayi sp. nov.

urn:lsid:zoobank.org:act:9DDFE14D-8E52-4F7D-B148-F552100865C9

Holotype. Museums and Art Galleries of the Northern Territory (hereafter, NTM) P5364, right epiplastron (Fig. 1).

Figure 1 Holotype of Chelodina (Chelodina) murrayi sp. nov., right epiplastron, NTM P5364.

(A) ventral view; (B) dorsal view; (C) anterior view; (D) posterolateral view; (E) interpretive drawing of A; (F) interpretive drawing of B. Arrow indicates ventrally curved marginal flange. Abbreviations: da, line of dermal attachment; GU, gular scute; HU, humeral scute; IN, intergular scute; PE, pectoral scute; sys, symphyseal surface. Hatched areas represent broken bone surfaces, grey areas represent adherent matrix. Scale bar = 20 mm.

Referred Specimens. NTM P5337, peripheral, probably left peripheral 2; NTM P5369, left hyoplastron; NTM P5370, nuchal; NTM 5371, left peripheral 8; NTM 5373, peripheral, probably left peripheral 10; NTM P5374, right xiphiplastron; NTM P5375, left hypoplastron; NTM P5376, mid-series costal; NTM P5377, proximal end of right costal 7; NTM P5378, indeterminate peripheral; NTM P5409, distal end of right costal 2; NTM P9810, left hypoplastron; NTM P9892, right hypoplastron; plus many other poorly informative shell fragments that are unregistered in the NTM palaeontology collection.

Locality and Horizon. All specimens come from the lower bone bed of the Waite Formation on Alcoota Scientific Reserve, 200 km northeast of Alice Springs, Northern Territory, Australia. The lower bone bed (Alcoota Local Fauna) is Late Miocene in age or Waitean in terms of Australian Land Mammal Ages (Megirian et al., 2010).

Etymology. In recognition of Peter Murray, who has led two decades of excavation at Alcoota and has published many significant contributions to our knowledge of the site.

Diagnosis. A species of C. (Chelodina) with the following autapomorphies: a sharp, ventrally-curved lip along the anterior margin of the gular scute of the plastron. Ventral side of the cervical scute of the carapace is 55% of the width of the dorsal side. Dorsally-curved lateral margins of the peripheral extending away from the bridge area both anteriorly and posteriorly, perhaps as far as peripheral 2 anteriorly and peripheral 10 posteriorly.

Differential Diagnosis. Apart from the diagnostic autapomorphies mentioned above C. (C.) murrayi has a unique combination of characters that allow it to be distinguished from all other Australasian chelids. It can be distinguished from all short-necked Australasian chelids by the presence of a midline contact between the gular scutes, anterior to the intergular scute on the plastron. It can be distinguished from Chelodina (Macrodiremys) colliei by the lateral expansion of the anterior lobe of the plastron, the location of the triple junction between the intergular, humeral and pectoral scutes on the epiplastron and a robust, medially extended anterior bridge strut of the hyoplastron. It can be distinguished from species of Chelodina (Macrochelodina) by the lateral expansion of the anterior lobe of the plastron, a broad, tear-drop shaped sutural surface on the anterior bridge strut, absence of a well-defined line on the anterior bridge strut demarcating the attachment of the dermis and the short junction of the pectoral scute pair that was exceeded in length by the intergular scute. Within Chelodina (Chelodina) it can be distinguished from C. (C.) longicollis and C. (C.) steindachneri by a large anterior bridge strut that almost certainly extended onto the pleural bones. The strongly dorsally curved margins of the peripheral bones distinguish it from C. (C.) steindachneri, C. (C.) pritchardi and C. (C.) reimanni which have no marginal curving. The well-developed and extensive nature of this curving also serves to distinguish the new species from C. (C.) novaeguineae and C. (C.) canni which only develop weak marginal curving in the bridge region. It can be distinguished from C. C. (C.) pritchardi, (C.) novaguineae, C. (C.) mccordi, and C. (C.) reimanni by the widening of the anterior plastral lobe anterior to the axial notch. The lack of a distinct lateral bulge of the femoral scute on the xiphiplastron distinguishes it from C. (C.) longicollis, C. (C.) novaeguineae and C. (C.) reimanni. Finally the failure of the intergular scute to extend broadly onto the hyoplastra distinguishes this species from C. (C.) canni.

Description

Carapace

Nuchal bone. The nuchal bone (NTM P5370) is the most informative part of the carapace that has been recovered from Alcoota (Fig. 2, Table 1). It is a roughly trapezoidal, bilaterally symmetrical plate that is only marginally longer than it is wide at its posterior end. The lateral edges that contacted the first peripherals on each side are gently concave whereas the posterior margin that would have sutured with the first pair of costals is convex. The seams that mark the boundaries of the scutes on the dorsal surface of the nuchal indicate that the anterior end was covered by a central cervical scute which was flanked on each side by the first pair of marginal scutes. The cervical scute is broadly rectangular and is 1.27 times longer than it is wide. Posteriorly the dorsal surface of the nuchal bone was covered entirely by a large first vertebral scute. The dorsal profile of the nuchal bone is gently concave. Ventrally the fine reticulate ornament and scute seams extend posteriorly for 20 mm before reaching the crescent-shaped scar that marks the attachment of the skin to the ventral surface of the carapace. The portion of the cervical scute that continues onto the ventral surface of the carapace is strongly narrowed relative to the dorsal portion. Its width is 51 per cent of the dorsal portion of the scute. The anterior rim of the nuchal is not curved dorsally, unlike the margins of all the known peripherals.

Figure 2 Chelodina (Chelodina) murrayi sp. nov., nuchal bone, NTM P5370.

(A) dorsal view; (B) ventral view; (C) interpretive drawing of A; (D) interpretive drawing of B. Abbreviations: CE, cervical scute; da, line of the dermal attachment; M1, first marginal scute; ssc1, sutural surface for first costal bone; ssp1, sutural surface for first peripheral bone; V1, first vertebral scute. Hatched areas represent broken bone surfaces. Scale bar = 20 mm.

Table 1 Measurements of selected elements of Chelodina (Chelodina) murrayi n. sp.

Element	NTM	ML
(mm)	AW
(mm)	PW
(mm)	MS
(mm)	HpS
(mm)	HS
(mm)	XS
(mm)	MW
(mm)	
Nuchal	P5370	57.3	32.0	52.0	—	—	—	—	—	
Epiplastron	P5364	—	—	—	(23.9)	—				
Hyoplastron	P5369	—	—	—	41.3	71.9	—	—	—	
Hypoplastron	P9810	—	—	—	33.6		(45.6)	(27.7)		
	P5375	—	—	—	(26.1)	—	(49.8)	—	—	
Xiphiplastron	P5374	—	—	—	(34.4)	(34.3)				
Peripheral 2	P5337	—	—	—	—	—	—	—	22.0	
Peripheral 10	P5373	—	—	—	—	—	—	—	24.0	
Notes.

ML midline length

AW anterior width

PW posterior width

MS length midline symphysis

HpS length of suture with hypoplastron

HS length of suture with hyoplastron

XS length of suture with xiphiplastron

MW marginal width

Costal bones. There are several costal bone fragments, of which NTM P5377, the proximal end of a right costal 7, is the most informative (Figs. 3A–3D). The costal bone extends all the way to the midline indicating that there was no exposed neural bone on the midline. Other fragments from the medial ends of costals also lack an exposed neural bone, indicating that there were at most one or two discontinuous exposed neurals, if any at all. Dorsally the bone is traversed by a single longitudinal seam that in life divided the fourth pleural scute from the fourth or fifth vertebral scute. The dorsal surfaces of all of the costal bones are rugose with rounded irregular ridges that tend to become elongated and transversely aligned toward the distal end (Fig. 3E). These replace the rounded flat tubercles separated by a network of finely etched grooves that are seen on the ventral surface of the plastron and the dorsal surface of some of the peripherals. A fossa with a sharp anterior rim excavates the ventral surface along the posterior margin, near the medial end. This represents the anterior part of the iliac scar. The medial ends of this specimen and other costal bones do not curve ventrally indicating that a longitudinal median furrow was not present.

Figure 3 Chelodina (Chelodina) murrayi sp. nov., costal bones.

(A–D), proximal end of right costal 7, (NTM P5377). A, dorsal view; B, ventral view; CD; (E), distal end of right costal 2 (NTM P5409) in dorsal view; (F–I) interpretive drawings of A–D, respectively; (J) Interpretive drawing of E. Abbreviations: ia, iliac articulation surface; P1, P2, P4, first, second and fourth pleural scutes; r, medial end of rib; rg, rib gomphosis; ss, symphyseal surface; ssc6, sutural surface for sixth costal bone; sssp, sutural surface for suprapygal bone; V4, fourth vertebral scute; vps, vertebral-pleural scute seam. Hatched areas represent broken bone surfaces, grey areas represent adherent matrix. Scale bar = 20 mm.

Peripheral bones. All of the preserved peripheral bones (NTM P5337, P5371, 5373 and P5378) have dorsally curved margins. The anterior-most preserved peripheral (NTM P5337) is almost certainly peripheral 2 (Figs. 4A–4F), while the posterior-most one (NTM P5373) is probably a peripheral 10 based on the sinuous posterior sutural line that would have contacted the pygal (Figs. 4G–4L). These indicated that the upcurved margins of the shell extended well anterior and posterior of the bridge. The margin of peripheral 2 is particularly strongly curved, similar to the degree displayed by peripherals from the bridge region of C. (C.) longicollis even though this specimen is clearly not from the bridge (Fig. 4B). The dorsal surface of the peripherals is ornamented with fine irregular rugae (e.g., NTM P5337, Fig. 4A) or a fine reticulate ornament of the same type seen on the surface of the bones of the plastron (e.g., NTM P5373, Fig. 4G).

Figure 4 Chelodina (Chelodina) murrayi sp. nov., peripheral bones.

(A–F), left second peripheral, (NTM P5337). A, dorsal view; B, ventral view; C, posterior view; D, interpretive drawing of A; E, interpretive drawing of B; F, interpretive drawing of C; (G–L) left tenth peripheral, (NTM P5373). G, dorsal view; H, ventral view; I, anterior view; J, interpretive drawing of G; K, interpretive drawing of H; L, interpretive drawing of I. Abbreviations: da, line of the dermal attachment; M2, M3, M10, M11, second, third, tenth and eleventh marginal scutes; P1, P4, first and fourth pleural scutes; ssc1, ssc8, sutural surfaces for first and eighth costal bones; sspe3, sspe9, sspe11, sutural surfaces for third, ninth and eleventh peripheral bones. Scale bar = 20 mm.

Plastron

Like the carapace the plastron is only known from isolated elements. However, unlike the carapace, enough pieces are known from similar sized individuals to produce a reconstruction (Fig. 5).

Figure 5 Chelodina (Chelodina) murrayi sp. nov., reconstruction of plastron in ventral view.

Abbreviations: AB, abdominal scute; AN, anal scute; ent, entoplastron; epip, epiplastron; FE, femoral scute; GU, gular scute; hyo, hyoplastron; hypo, hypoplastron; HU, humeral scute; IN, intergular scute; PE, pectoral scute; xip, xiphiplastron. Grey areas represent areas of missing bone that have been reconstructed. Scale bar = 50 mm.

Epiplastron. The epiplastron is represented by the holotype, NTM P5364 (Fig. 1, Table 1). It is a roughly elliptical plate with a rounded convex anterolateral margin. The margin forms a thin sharp edge that is weakly reflexed ventrally at its posterior end where it forms the anterolateral margin of the humeral scute. The ventral curvature of the margin becomes more strongly developed anteriorly, along the anterolateral margin of the gular scute. Anteriorly the ventral surface is crossed by the seam between the gular and intergular scutes. This seam reaches the median symphyseal line posterior to the anterior margin indicating that the intergular scute was retracted from the anterior margin and the two gular scutes had an anterior midline contact. However this midline contact appears to have been rather short, with a pointed anterior process of the intergular scute intruding deeply between the pair. The posterior end of the epiplastron is damaged but a seam line can be seen clearly curving from the lateral margin of the intergular scute toward the lateral margin of the epiplastron. This seam forms the posterior border of the humeral scute and indicates that the pectoral scute extended onto the posterior surface of the epiplastron. The humeral-pectoral scute seam meets the humeral-intergular scute seam at the medial margin of NTM P5364, but this is clearly a broken edge, not the sutural surface that contacts the entoplastron. Therefore the triple junction of the humeral, pectoral and intergular scute occurs on the epiplastron.

Hyoplastron. The complete hyoplastron (NTM P5369) is the largest single piece of turtle shell recovered from Alcoota (Fig. 6, Table 1). It represents a moderately small turtle with a plastron width at the level of the axial notches of about 105 mm. The anterior lobe expands slightly laterally, anterior to the axial notch as in adult C. (C.) canni (McCord & Thomson, 2002), though not to the extreme displayed by C. (C.) longicollis (e.g., NTM R27168). In lateral view the anterior lobe is slightly upcurved. Its margin is a simple edge that is not ventrally reflexed as in the margin of the epiplastron. A strongly developed seam representing the junction of the abdominal and pectoral scutes extends transversely across the ventral surface, approximately one quarter of the length of the bone from its posterior margin. The seam curves posteriorly as it approaches the medial symphyseal suture. There is no seam present near the anteromedial corner of the bone where the medial symphysis meets the suture between the hyoplastron and the entoplastron. This indicates that the posterior end of the intergular scute terminated on the entoplastron and did not extend onto the hyoplastra as it does in C. (C.) canni (pers. obs., NTM R16325). Anteriorly the hyolplastron forms a short jagged sutural surface that contacted the epiplastron. Lateral to the triangular notch in this suture, the ventral surface is marked by a seam that extends from the hyoplastron-epiplastron suture to the lateral margin of the hyoplastron. This indicates that the posterolateral corner of the humeral scute extended onto the hyoplastron. Because the humeral-pectoral scute seam intersects the hyoplastron-epiplastron suture rather than extending transversely to the hyoplastron-entoplastron suture, the pectoral scute must have extended onto the epiplastron, as is shown by the epiplastron from Alcoota (NTM P5364). The axial notch extends for slightly less than half the length of the hyoplastron. The bridge region, posterior and lateral to the axial notch is set at a very broad angle to the ventral plate of the hypolastron. Although some crushing may have further flattened the hypolastron it seems likely that the shell was quite dorsoventrally shallow in life. The anterior end of the bridge region curves sharply dorsally and medially to form the anterior bridge strut. No distinct line can be seen on this surface marking the attachment of the dermis. A rounded notch that is 1.7 mm across excavates the anterodorsal margin of the bridge at the point that it curves medially to form the anterior bridge strut (Fig. 7C). This notch would have formed a complete foramen when articulated with the third peripheral. Extant Chelodina have a well developed foramen in the same position to allow for the passage of the axillary duct of Rathke’s gland (Goode, 1967; Weldon & Gaffney, 1998). The sutural surface of the anterior bridge strut that contacted the carapace is tear-drop shaped with a strongly expanded medial end. The medial projection of the anterior bridge strut is great enough to be confident that it would have made broad contact with the ventral surface with the first costal bone when articulated with the carapace.

Figure 6 Chelodina (Chelodina) murrayi sp. nov., left hyoplastron, NTM P5369.

(A) dorsal view; (B) ventral view; (C) anterior view; (D) lateral view; (E) interpretive drawing of A; (F) interpretive drawing of B; (G) interpretive drawing of C; (H) interpretive drawing of D. Abbreviations: AB, abdominal scute; abs, anterior bridge strut; HU, humeral scute; PE, pectoral scute; ssca, sutural surface for articulation with the carapace; ssen, sutural surface of the entoplastron; ssep, sutural surface of the epiplastron; sshy, sutural surface of the hypoplastron; sys, symphyseal surface. Scale bar = 20 mm.

Figure 7 Anterior bridge struts of various chelodinines in oblique, anterior-ventral-lateral view.

Top row: whole specimens with area of enlargement indicated by a box. Bottom row: enlargement of anterior bridge strut area. (A) Emydura sp., NTM unregistered comparative collection; (B) Chelodina (Chelodina) longicollis; (C) Chelodina (Chelodina) murrayi. Dotted line indicates likely extent of foramen for the axillary duct of Rathke’s gland. Note that sediment partly infills this notch. Abbreviations: da, line of dermal attachment; rgf, foramen for the axillary duct of Rathke’s gland.

Hypoplastron. The hypoplastron is represented by three specimens, all of which are damaged. The main body is a roughly square plate with a laterally protruding bridge process (Figs. 8 and 9). The ventral surface of the main body is traversed by a straight transverse seam, dividing the abdominal scute from the femoral scute. The lateral end of the seam terminates at the rounded inguinal notch between the body and the bridge process. Posterior to the inguinal notch the lateral margin of the femoral scute is produced laterally to form a modest semilunate flange. As in the hyoplastron the bridge is set at a very broad angle to the main body, indicating a markedly shallow shell not unlike C. (C.) steindachneri.

Figure 8 Chelodina (Chelodina) murrayi sp. nov., left hypoplastron, NTM P5375.

(A) dorsal (internal) view; (B) ventral (external) view; (C) Interpretive drawing of A; (D) Interpretive drawing of B. Abbreviations: AB, abdominal scute; b, bridge; FE, femoral scute; ssh, sutural surface for the hyoplastron; ssx, sutural surface for the xiphiplastron; sys, symphyseal surface. Hatched areas represent broken bone surfaces. Scale bar = 20 mm.

Figure 9 Chelodina (Chelodina) murrayi sp. nov., left hypoplastron, NTM P9810.

(A) dorsal (internal) view; (B) ventral (external) view; (C) Interpretive drawing of A; (D) Interpretive drawing of B. Abbreviations: AB, abdominal scute; b, bridge; FE, femoral scute; ssh, sutural surface for the hyoplastron; ssx, sutural surface for the xiphiplastron; sys, medial symphyseal surface. Hatched areas represent broken bone surfaces, grey areas represent patches of adherent matrix. Scale bar = 20 mm.

Xiphiplastron. The single complete xiphiplastron (NTM P5374) is a flat trapezoidal plate (Fig. 10, Table 1). The ventral surface was largely covered by the anal scute with a narrow band of the femoral scute covering the anterior end of the bone. The lateral margin is damaged in the middle but the posterior and anterior ends of the lateral margin are complete. These show that the lateral margin in the region of the femoral scute was in line with the posterior lateral margin of the anal scute and did not project laterally to form a femoral bulge like some chelodinines such as Chelodina (Chelodina) longicollis and Birlimarr gaffneyi (Megirian & Murray, 1999). The posterolateral corner is moderately thickened and upturned. It is not extended posteriorly and the lateral and posterior margins meet to form a right-angled corner. Mirror-imaging (Fig. 10E) indicates that the anal notch was triangular. The dorsal surface bears sutural scars for the ischium and pubis. The pubic scar is centrally located in the anterior half of the xiphiplastron and is roughly oval and oriented with its long axis extending obliquely anteromedial to posterolateral. The ischiadic scar is elongate and tongue-shaped and extends to the midline suture, where it would have met its partner and formed a continuous crescentic scar. The anterior and posterior margins of the scar are parallel and do not diverge laterally. The posterolateral margin is rounded.

Figure 10 Chelodina (Chelodina) murrayi sp. nov., right xiphiplastron, NTM P5374.

(A) ventral view; (B) dorsal view; (C) interpretive drawing of A; (D) interpretive drawing of B; (E) reconstruction of articulated xiphiplastron pair in dorsal view. Abbreviations: AN, anal scute; FE, femoral scute; sshy, sutural surface for hypoplastron; ssi, sutural surface for ischium; ssp, sutural surface for pubis; sys, symphyseal surface; tm, tooth mark. Hatched areas represent broken bone surfaces, grey areas represent areas of adherent matrix. Scale bar = 20 mm.

Discussion

Phylogenetic relationships

The midline contact of the gular scutes anterior to the intergular is a well-known synapomorphy of Chelodina (Gaffney, 1977; Gaffney, 1981; Georges & Thomson, 2010). The rounded posterolateral margin of the ischiadic scar on the xiphiplastron is a further synapomorphy of Chelodina (Lapparent de Broin & Molnar, 2001). Finally the enlargement of the axillary scent gland foramen would appear to be also a synapomorphy of Chelodina. The foramen is large in Chelodina (Chelodina) (e.g., diameter of 2.2 mm in C. (C.) longicollis; NTM R27168) and Chelodina (Macrochelodina) (e.g., C. (M.) expansa; Goode, 1967, Fig. 134) whereas it is comparatively tiny in members of the short-necked clade (e.g., diameter of 0.9 mm in Emydura sp.; NTM unregistered) (Fig. 7A).

Lateral widening of the anterior lobe of the plastron is a character that has only been observed among members of the nominal subgenus (e.g., C. (C.) longicollis and C. (C.) canni) and is probably a synapomorphy of the subgenus that reverses in some members of the C. (C.) novaeguineae species complex. Further supporting a position within C. (Chelodina), the length of the intergular scute was almost certainly greater than twice the length of the seam between the pectoral scutes. In short-necked chelodinines the ratio of the length of the intergular scute to the length of the shared pectoral seam ranges from 0.55 (Emydura sp.) to 1.35 (Pseudemydura umbrina) a range that overlaps with the range of 1.14 to 1.53 seen in Chelodina (Macrochelodina) (Table 2). C. (Macrodiremys) colliei also falls in this range with a ratio of 1.43. Thus a ratio of less than 2 is clearly primitive for Chelodininae. In contrast all members of C. (Chelodina) have an intergular scute that is more than twice the length of the shared pectoral seam and can range up to 5.35 times longer in C. (C.) canni (Table 2).

Table 2 Ratio of intergular scute length to length of the shared seam between pectoral scutes in various chelodinines.

Species	Ratio	Source	
Pseudemydura umbrina	1.35	Burbidge, Kirsch & Main, 1974, Fig. 1	
Emydura sp.	0.55	NTM (MCA unregistered)	
Chelodina (Macrodiremys) collei	1.43	Burbidge, Kirsch & Main, 1974, Fig. 2	
Chelodina (Macrochelodina) rugosa	1.53	NTM R24814	
Chelodina (Macrochelodina) rugosa	1.40	Thomson, Kennett & Georges, 2000, Fig. 6B	
Chelodina (Macrochelodina) burrungandjii	1.23	NTM R35010	
Chelodina (Macrochelodina) burrungandjii	1.14	Thomson, Kennett & Georges, 2000, Fig. 6A	
Chelodina (Macrochelodina) parkeri	1.18	Rhodin & Mittermeier, 1976, Fig. 15	
Chelodina (Chelodina) longicollis	3.25	NTM R27168	
Chelodina (Chelodina) canni	5.35	NTM R16325	
Chelodina (Chelodina) novaeguineae	3.20	McCord & Thomson, 2002, Fig. 3F	
Chelodina (Chelodina) novaeguineae	2.79	Rhodin, 1994a, Fig. 7	
Chelodina (Chelodina) pritchardi	3.13	Rhodin, 1994a, Fig. 1	
Chelodina (Chelodina) reimanni	2.37	Philippen & Grossmann, 1990, Fig. 2	

The state of this character can be determined in C. (Chelodina) murrayi despite the fragmentary nature of the material. The length of the junction between the pectoral scutes on the hyoplastron (P5369) is 28 mm. Because the intergular scute extends close to the posterior end of the entoplastron in all Chelodina, the total length of the pectoral scute junction in life could not have been more than a few millimetres longer than the length of the seam present on the hyoplastron. The depth of the notch in the anterior margin of the hyoplastron for receiving the entoplastron is 27 mm. Thus the midline length of the intergular scute was approximately 27 mm plus an unknown length that protruded anteriorly between the epiplastra. There is 18 mm of intergular scute overlapping the midline suture of the epiplastron. Because the width of this epiplastron from the midline to the lateral edge is a good match for the width of the anterior end of the hyoplastron (Fig. 6A) they must derive from similar sized individuals or possibly even the same individual. Thus we can be confident that the length of the intergular in an individual the size of P5369 or P 5364 was 45 mm (18 + 27) plus an unknown length between the anterior end of the hyoplastron and the posterior end of the midline suture between the epiplastra. Thus, even when the unknown segment of the intergular is ignored, the ratio of intergular to the shared pectoral seam exceeds the ratio of 1.5 which lies at the upper end of the range in Chelodina (Macrochelodina) and Chelodina (Macrodiremys). Indeed the reconstruction suggests that the total length of the intergular was likely to be around 65 mm, which is 2.3 times the length of the shared seam between the pair of pectoral scutes.

Finally supporting the subgeneric placement of C. (C.) murrayi is the absence of a distinct line demarcating the attachment of the dermis on the anterior surface of the anterior bridge strut. In short-necked chelodinines (Fig. 7A) and in Chelodina (Macrochelodina) ssp. (A Yates, personal observations, NTM R35010, NTM R24813) there is a distinct, sharp line extending anterodorsally from the axillary notch to the peripherals of the carapace that indicates the position that the dermis attached to the shell and divides the ornamented external surface from the smooth inner surface. In Chelodina (Chelodina) however there is no distinct line and the external surface merges gradually with the internal surface (A Yates, personal observations, NTM R16325, R27168; Fig. 7B). The latter condition would appear to be the derived one and is a further synapomorphy of Chelodina (Chelodina).

Within Chelodina (Chelodina) the large medially inflected and terminally expanded anterior bridge strut of the hyoplastron indicate that its relationships lie with the C. (C.) novaeguineae species complex (McCord & Thomson, 2002). McCord & Thomson (2002) included C. (C.) novaeguineae, C. (C.) canni, C. (C.) reimanni and C. (C.) mccordi in this complex. I also include C. (C.) pritchardi because it has a large anterior bridge strut that contacts the first costal (Rhodin, 1994a) and is more closely related to other members of the complex than it is to either C. (C.) longicollis or C. (C.) steindachneri (Georges, Adams & McCord, 2002). The condition in basal species of the subgenus, i.e., (C. (C.) longicollis and C. (C.) steindachneri, is to have reduced anterior bridge struts that do not extend far medially and fail to contact the costal bones (Thomson, 2000). The same condition obtains in C. (Macrodiremys) colliei (Thomson, 2000) which is the sister group to C. (Chelodina), indicating that reduced anterior bridge struts are primitive condition for the subgenus. Thus the C. (C.) novaeguineae species complex have apparently reversed this condition and have medially extensive anterior bridge struts that broadly contact the costal bones like those of Chelodina (Macrochelodina) and short-necked chelodinines (Thomson, White & Georges, 1997; McCord & Thomson, 2002).

Unity of the hypodigm

It is obvious that much of the diagnosis and description of the new taxon is reliant upon the referral of several unassociated specimens to a single taxon. It should be noted that even if there was more than one turtle taxon present at Alcoota, Chelodina (Chelodina) murrayi would still stand as a valid taxon because the holotype epiplastron displays an autapomorphy as well as a synapomorphy of Chelodina. Nevertheless the referral of the Alcoota turtle sample to a single species is the most likely hypothesis based on the following observations.

Firstly multiple elements (the epiplastron, hyoplastron and the xiphiplastron) display synapomorphies of Chelodina: anteriorly enclosed intergular scute (Fig. 1A), enlarged foramen for Rathke’s gland (Fig. 7C), rounded posterolateral margin of ischiadic scar (Figs. 10B and 10D). The hyoplastron also shows additional synapomorphies of Chelodina (Chelodina): lateral expansion of anterior lobe of plastron, absence of a distinct line of attachment of the dermis on the anterior bridge strut and a short pectoral-pectoral scute seam.

Other elements display plesiomorphies that are only present in Chelodina or Pseudemydura among Chelodininae. These plesiomorphies include the extension of the pectoral scute onto the posterior end of the epiplastron, a triple junction between the humeral, pectoral and intergular scute located on the epiplastron and a skin-carapace contact on the nuchal bone that lies posterior to the anterior margin of the carapace. None of the preserved elements show any characters of the distinctive Pseudemydura, thus these plesiomorphies add further support to the conclusion that most, if not all, of the Alcoota chelid specimens belong to Chelodina.

None of the preserved elements show any characters that are inconsistent with referral to Chelodina (Chelodina). These observations strongly suggest that only Chelodina (Chelodina) is present in the Alcoota assemblage. Extant Chelodina (Chelodina) species have almost exclusively allopatric ranges (Kennet et al., 1992; Rhodin, 1994a; Rhodin, 1994b; McCord & Thomson, 2002; Georges & Thomson, 2010). The Fitzroy-Dawson drainage includes hybrids between C. (C.) canni and C. (C.) longicollis and would appear to represent the sole river system where two species of Chelodina (Chelodina) overlap (Georges, Adams & McCord, 2002). Thus it is very unlikely that more than one species of Chelodina (Chelodina) is present in the Alcoota local fauna and the hypodigm of C. (C.) murrayi can be safely treated as pertaining to a single species.

Biogeography and evolution of Chelodina (Chelodina)

Extant members of the C. (C.) novaeguineae complex are restricted to the far north of Australia (C. (C.) canni), New Guinea (C. (C.) novaeguineae, C. (C.) reimanni and C. (C.) pritchardi) and Indonesia (C. (C.) mccordi). The presence of C. (C.) murrayi at Alcoota indicates that this clade extended further south in the past. At present it is not possible to determine if C. (C.) murrayi is an unusual southerly extension of an otherwise tropical clade, or whether the species group originated south of its present range and only later radiated in the tropics to the north of continental Australia.

The presence of a member of the C. (C.) novaeguineae species complex in the late Miocene indicates that stem members of C. (C.) steindachneri and C. (C.) longicollis were in existence at this time and their divergence was not related to climatic fluctuations of the Pleistocene as has been suggested (Kennet et al., 1992).

Palaeobiology

Among chelodinines it is the members of Chelodina (Chelodina) that are best able to cope with unpredictable drying events. C. (C.) steindachneri occupies some of the most arid habitats of any chelodinine, where surface water is ephemeral and infrequent (Cann, 1998). C. (C.) longicollis and C. (C.) canni are reported to travel many kilometres over land (Stott, 1987; Covacevich et al., 1990) and are capable of prolonged aestivation (Bill Cook, reported in Kennet et al., 1992; Roe & Georges, 2007).

It is therefore unsurprising to find that the Alcoota chelid belongs to Chelodina (Chelodina) because palaeoenvironmental indicators suggest that surface water in the Waite Basin, during the late Miocene, was prone to episodic drying events. It is notable that the Alcoota assemblage does not contain any fish. While a preservational bias may account for the lack of smaller fish bones it does not explain the absence of robust lungfish toothplates which are generally abundant in freshwater deposits throughout central and eastern Australia from the late Oligocene through to the Pleistocene (Kemp, 1991; Kemp, 1993). The lack of lungfish toothplates therefore indicates that surface water at the Alcoota site was not permanent. Other aquatic taxa are scarce. Chelodina (Chelodina) murrayi itself requires a minimum of just two individuals to account for its known remains. Crocodiles are present, but these are massive, altirostral, semiziphodont mekosuchins of the genus Baru (A Yates, personal observations) with clear indications that they specialised on large vertebrate prey (Willis, Murray & Megirian, 1990) and thus may not have been as strongly tied to deep permanent water as extant crocodylids are.

Conclusions

The chelid material from the late Miocene Alcoota Local Fauna of central Australia can be referred to a single species that is here named Chelodina (Chelodina) murrayi. Within the subgenus, the affinities of the new species lie with the C. (C.) novaeguineae species complex which now occurs in tropical habitats to the north of Alcoota (mostly outside continental Australia). Thus both the subgenus, and the C. (C.) novaeguineae species complex were more widespread in the past and diversification of the extant species began prior to the late Miocene.

This work would not have been possible without the immense effort put into running the Alcoota field program for over two decades by P Murray and D Megirian. Many people have participated in these excavations over the years but J Archibald stands out as the collector of the holotype and several other important pieces of Chelodina (Chelodina) murrayi. Access to comparative specimens was facilitated by G Dally and S Horner. S Thomson alerted me to the nomenclatural complications surrounding Chelodininae and its primary author.

Additional Information and Declarations

Competing Interests

Author Contributions

New Species Registration

The author declares he has no competing interests.

Adam M. Yates analyzed the data, contributed reagents/materials/analysis tools, wrote the paper.

The following information was supplied regarding the registration of a newly described species:

Zoobank

urn:lsid:zoobank.org:act:9DDFE14D-8E52-4F7D-B148-F552100865C9.

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
