# Peer review of "A new species of long-necked turtle (Pleurodira: Chelidae: Chelodina) from the late Miocene Alcoota Local Fauna, Northern Territory, Australia"

_PeerJ, doi:10.7717/peerj.170_

## Round 0.1 · original submission · Minor Revisions

The reviewers were uniformly positive about your paper, and I agree with their assessment. Only two specific changes are needed, based on the reviewer comments as well as my own reading of the paper:

1) Please update the nomenclature per the suggestions of Reviewer 2.

2) Regarding the purported bite marks, please provide a more detailed description, and also consider the possibility that they may be non-traumatic lesions as described elsewhere in the literature (detailed below). Alternatively, if you are not satisfied with an identification of these features as bite marks in the end, you may consider removing this section of the paper. In my opinion, the manuscript would not be harmed by doing so.

OTHER COMMENTS FROM THE EDITOR:
- Please consider including measurements (lengths, widths, thicknesses) for all complete elements within the description.

- Could you include just a little more description of the purported tooth marks? Size, spacing, morphology, etc.? A recent paper by Boyd et al. (2013) discussed identification of feeding traces from crocodilians in some detail. Are there signs of genuine bone breakage/indentation? Or could these markings represent the random pits found in many extant turtle shells, even from geographic areas without large and toothy aquatic predators (see Rothschild et al. 2013 for a discussion)? Is Baru the only crocodilian known from the formation? In short, I am not entirely convinced that the markings you describe are necessarily bite marks, at least as presented in the text. Because this is such a minor part of the text, you may wish to just delete it.

CITATIONS
Boyd CA, Drumheller SK, Gates TA (2013) Crocodyliform Feeding Traces on Juvenile Ornithischian Dinosaurs from the Upper Cretaceous (Campanian) Kaiparowits Formation, Utah. PLoS ONE 8(2): e57605. doi:10.1371/journal.pone.0057605

Rothschild, BM, H-P Schultze, R. Pellegrini. 2013. Osseous and Other Hard Tissue Pathologies in Turtles and Abnormalities of Mineral Deposition. In: Morphology and Evolution of Turtles, pp. 501-534.

·

Basic reporting

No Comments

Experimental design

No comments

Validity of the findings

No comments

Additional comments

A welcome review of the knowledge that has accumulated to date of the fossil history of this genus of turtles, and one that adds several characters that will assiste future workers on the group.

·

Basic reporting

No Comments

Experimental design

No Comments

Validity of the findings

No Comments

Additional comments

I have enjoyed this paper and extend my congratulations on defining this species. I need to make a couple of corrections on the nomenclature. I note you have largely followed myself and Arthur Georges (2010) on this and will admit that our nomenclature for the Chelidae was slightly wrong. This has been published so I will summarize what was wrong and give you the relevant references.

The higher order nomenclature of the Chelidae was originally proposed by Baur in 1893. I recently discovered this myself and hence Georges et al. 1998 cannot have credit for it. In the most recent IUCN checklist (van Dijk et al., 2012) the annotations for the Chelidae correct the nomenclature. I suggest you adjust your Systematic Palaeontology section, lines 14 and 15, to be in line with the findings of that paper. Throughout your introduction you will also need to update and reword your discussions of subfamily Chelodininae to ensure the name is referred to Baur (1893). It is still correct that the reciprocal monophyly was proposed by Georges et al., 1998 and they did suggest the use of subfamilies. They just did not know about the previous attempt to do this by Baur, 1893.

Below are the higher orders with correct citations and full citations for Gray 1825 and Baur 1893. I also include the citation for the 2012 checklist where the nomenclature was determined and a link to the pdf of that paper.

Chelidae Gray 1825
Chelinae Gray 1825
Chelodininae Baur 1893
Hydromedusinae Baur 1893

Baur, Georg. 1893. Notes on the classification and taxonomy of the Testudinata. Proceedings of the American Philosophical Society 31:210–225.
Gray, John Edward. 1825. A synopsis of the genera of reptiles and amphibia, with a description of some new species. Annals of Philosophy (2)10:193–217.
van Dijk , P.P., Iverson, J.B., Shaffer, H.B., Bour, R., and Rhodin, A.G.J. 2012. Turtles of the world, 2012 update: annotated checklist of taxonomy, synonymy, distribution, and conservation status. Chelonian Research Monographs No. 5, pp. 000.243–000.328.

Checklist available here:
http://www.iucn-tftsg.org/wp-content/uploads/file/Accounts/crm_5_000_checklist_v5_2012.pdf

I do think you need to make these nomenclatural corrections, sorry that my 2010 paper also had it slightly wrong so I blame myself for that. Apart from that I think the paper should be accepted.

Just for your information, and this is unpublished, when I was at Naracourte many years ago there was a relatively complete shell of C. (Chelodina) probably longicollis in the collection at the University. From the Naracourte caves. As far as I am aware it has never been published though.

Feel free to email me if you would like to see the Baur, 1893 paper, I have a pdf of it.

---

## Round 0.2 · Minor Revisions

1) I note that the museum abbreviation "NTM" is not defined in the text (it is my fault for not catching it earlier!). I am assuming this refers to "Museums and Art Galleries of the Northern Territory," but it should be explicitly defined so it is easy for other researchers to know where the holotype is located.

2) In the previous version, I had requested some additional measurements for elements if possible. This was not incorporated into the manuscript, and not referenced in the rebuttal letter. If the specimens are not easily accessible, the manuscript can go as is, but otherwise some basic data (e.g., length along sutural surfaces) for a few complete elements (e.g. NTM P5370, P5364, P5374) would be important. I note that the width of NTM P5369 is included still from the original version.

Once these final issues are addressed, I should be able to issue a final decision in very short order.

---

## Round 0.3 · accepted · Accept

Thank you for your quick attention and flexibility on the last round of comments!